# Achieving integrated care through commissioning of primary care services in the English NHS: a qualitative analysis

Imelda McDermott,[1] Kath Checkland,[1] Valerie Moran,[2] Lynsey Warwick-Giles[1]

¹Centre for Primary Care, University of Manchester, Manchester, UK
²Luxembourg Institute of Health and Luxembourg Institute of Socio-Economic Research, Luxembourg, Luxembourg

**Correspondence to**
Dr Imelda McDermott;
imelda.mcdermott@manchester.ac.uk

## ABSTRACT

**Objectives** Since April 2015, Clinical Commissioning Groups (CCGs) have taken on the responsibility to commission primary care services. The aim of this paper is to analyse how CCGs have responded to this new responsibility and to identify challenges and factors that facilitated or inhibited achievement of integrated care systems.

**Design** We undertook an exploratory approach, combining data from interviews and national telephone surveys, with analysis of policy documents and case studies in four CCGs. Data were analysed using thematic content analysis.

**Setting/participants** We reviewed 147 CCG application documents and conducted two national telephone surveys with CCGs (n=49 and n=21). We interviewed 6 senior policymakers and 42 CCG staff who were involved in primary care co-commissioning (general practitioners and managers). We observed 74 primary care commissioning committee meetings and their subgroups (approx. 111 hours).

**Results** CCGs in our case studies focused their primary care commissioning activities on developing strategic plans, 'new' primary care initiatives, and dealing with legacy work. Many plans focused on incentivising and supporting practices to work together and provide a broad range of services. There was a clear focus on ensuring the sustainability of general practice. Our respondents expressed mixed views as to what new collaborative service models, such as the new models of care and sustainability and transformation partnerships (STPs), would mean for the future of primary care and the impact they could have on CCGs and their members.

**Conclusions** There is a disconnect between locally based primary care and the wider system. One of the major challenges we identified is the lack of knowledge and expertise in the field of primary care at STP level. While primary care commissioning by CCGs seems to be supporting local collaborations between practices, there is some way to go before this is translated into broader integration initiatives across wider footprints.

## INTRODUCTION

The quest to achieve 'integrated care' is a priority that has been identified internationally. Problems concerning the fragmentation of services between healthcare providers is an issue commonly discussed in many European countries.[1] Integration and coordination

## Strengths and limitations of this study

► The first study investigating the delegation of primary care commissioning responsibilities from National Health Service England to Clinical Commissioning Groups (CCGs).
► In-depth case studies of four CCGs nationally complemented with two national telephone surveys of CCGs.
► In-depth interviews with senior policymakers who played a role in the development of primary care commissioning policy.
► This study provides learning on how to integrate care vertically or 'knit' local primary care plans with regional plans.
► The study was undertaken in a rapidly developing policy environment therefore results represent a snap shot in time of a changing landscape.

are sought between primary and secondary care, curative and preventive (public health) healthcare, and between specialties within particular sectors.[2] Primary care, as the cornerstone in any health system, has been argued to have a central role in integrating care within a health system.[3] In countries with a general practitioner (GP) gatekeeping model, such as the UK and Denmark, this model has been used as an 'organisational mechanism' to promote integration.[4]

In the context of the English National Health Service (NHS), efforts to strengthen primary care have been pursued by active commissioning of primary care services.[5][6] This means the planning and managing of primary care services should involve contracting and procurement and clinically led service redesign and engagement with local stakeholders.[6] Hence contractual mechanisms and incentives used would have a significant impact on the way that services are delivered and therefore on the way they interact with other sectors.[7][8] The Health and Social Care Act 2012 (HSCA12) has created fragmentation in the system with commissioning responsibilities for

local populations now divided between several different organisations. Clinical Commissioning Groups (CCGs), which are groups of GP practices commission the majority of health services. However, the responsibility for commissioning of primary care services was initially given to a new arm's length body, NHS England (NHSE). This was to ensure a more standardised model and consistency in the management of primary care. However, in 2014, 2 years following the enactment of HSCA12, NHSE delegated this responsibility to CCGs. It was thought that this would enable CCGs to commission more integrated care pathways. The aim of this paper is to analyse this recent development and how the delegation of primary care commissioning responsibilities from NHSE to CCGs may contribute (or not) to achieving integrated care systems.

This paper provides a valuable contribution to the existing literature on commissioning. The commissioning of primary care services is an under-researched area, and this paper provides evidence on how commissioning is practised locally. This paper raises some fundamental issues relevant to the integration agenda in the current NHS in England.

We start by providing a brief history of the commissioning of primary care services in England, followed by the current policy context. Findings are presented in terms of the activities undertaken under primary care commissioning and how these activities relate to wider national initiatives to achieve integrated care. We conclude with a discussion of the challenges and factors facilitating these developments.

## HISTORY OF COMMISSIONING OF PRIMARY CARE SERVICES IN ENGLAND

The current primary care system in England, whereby GPs are contractors to the NHS, was born out of the decision made at the establishment of the NHS in 1947.[7] This enabled GPs to remain independent to the NHS, minimising their opposition to the NHS.[9 10] There was little planning for GP services from 1948 to 1990.[7] GPs were contracted as individuals and payment were governed by the number of registered patients and by the services provided. GP contracts were administered by executive councils (1948–1972) whose membership included heavy representations of GPs themselves.[11]

A unified system of administration was introduced in 1973[12] which integrated the planning and delivery of hospital services (administered by hospital boards), GP services (administered by executive councils) and personal health services (administered by local authorities such as maternity services, vaccination and ambulance services). A body called the Family Practitioner Committee (FPC) was created to replace the executive councils and its function was to administer the provision of primary care services.[12] Health authorities were responsible for the administration and coordination of primary care services, hospital services, community

services and services requiring collaboration with the local government.

The internal market was created in 1989[13] by the conservative government, introducing a split between the purchasers and providers of care, with a view to using competition between providers to achieve better 'value for money'. Purchasers would 'commission' health services from providers by entering into contracts to deliver an agreed volume of services at a price. Purchasing would be more than simply contracting with and paying for providers to supply health services; providers would be made to compete for resources to encourage greater efficiency, responsiveness and innovation. The FPC continued to hold GPs' contract and was responsible for monitoring expenditure against the budget. It is worth highlighting here that the term primary care commissioning is used in the literature to refer to both commissioning by primary care (also known as clinically led or primary care-led commissioning) and commissioning *of* primary care services. In this paper, the term is used to refer to the latter and it means commissioning of services provided by GP s services).

In spite of a rhetorical commitment to competition, payments of GP practices continued to be governed by a set of rules, with little local control over service development or provision. The notion of active commissioning started to gain prominence when the new labour government came to power in 1997. Responsibility for commissioning all types of services for a geographical population was given to newly established primary care trusts (PCTs), who were encouraged to start using a wider variety of contractual mechanisms to encourage new entrants into the primary care system. Traditionally, GP practices held a general medical services (GMS) contract, a nationally directed contract between NHSE and a practice, or a personal medical services (PMS) contract, a local contract agreed between NHSE and the practice. During the 2000s, PCTs were encouraged to procure new primary care services using a new contractual form, the alternative providers of medical services (APMS) contract. Crucially, such contracts could be held by non-traditional service providers, including private companies, and they could be adjusted to specify a different range of services. PCTs were thus encouraged to actively shape the supply of services in their areas, introducing competition and actively procuring services to meet population needs.

## CURRENT POLICY CONTEXT

The HSCA12 gave responsibility for commissioning primary care services to a new national body, NHSE. However, it proved difficult for this national body to actively commission services to meet local needs,[7] and in May 2014, following the appointment of Simon Stevens as the chief executive of NHSE, CCGs were given 'new powers' to take on a greater role in commissioning of primary care services (known as primary care co-commissioning). This would enable CCGs to take on a more

integrated approach and 'unlock the full potential of their (CCGs') statutory duty'.[14] Although NHSE is responsible for all the four primary care services (medical, dental, eye health and pharmacy), the delegation of responsibility focused on primary medical care.

The vision and aims of co-commissioning were described in relation to the wider agenda set out in the NHS Five Year Forward View (FYFV),[15] which set out a broad consensus on what the future of the NHS needed to be. The FYFV emphasised the need to develop more integrated care providers or networks of care providers to meet the needs of local people, especially those with long-term conditions and multiple health problems. Three types of new care models were piloted in March 2015: multispecialty community providers (blending primary and specialist services in one organisation and multidisciplinary teams providing services in the community); primary and acute care systems (integrating primary, hospital and mental health services) and enhanced health in care homes (multiagency support and the use of new technologies to help people stay at home).

To deliver the FYFV, all local organisations are required to work together across defined geographical footprints to produce a 5-year sustainability and transformation plan (later partnership) (STP).[16] STPs are place-based plans detailing how commissioners and providers would work together collaboratively to deliver more integrated care. The geographical scope of each STP was locally defined, based on natural communities, existing working relationships and patient flows, taking account of the scale needed to deliver the services. There are 44 STPs, each with a lead appointed by NHSE. STP leads are responsible for overseeing regional planning across the health and care system, including the reconciliation of different, often competing, interests of organisations to meet the needs of the local population. The most advanced STPs are expected to evolve into integrated (formerly accountable) care systems, which will be given more autonomy over the local health system including delegated commissioning powers for primary care and specialised services, devolved transformation funding and streamlined regulatory arrangements.

Co-commissioning is seen as the beginning of a journey towards place-based commissioning, defined as different commissioners coming together to jointly agree commissioning strategies and plans, using pooled funds, for services for a local population.[14] It is seen as an 'organisational mechanism'[4] to achieve integrated care. Co-commissioning is intended to enable CCGs to shift resources between sectors, hence dissolving traditional boundaries and supporting integrated care.[14] The 'new powers'[17] given to CCGs under co-commissioning includes: designing, monitoring, negotiating and terminating core general medical services contracts (GMS, PMS and alternative providers of medical services contracts); designing local incentive schemes as an alternative to the QOF; making decisions on whether to establish new GP practices in an area and approving practice mergers. CCGs

will thus play a crucial role in supporting the development of integrated services in which primary care services complement and link seamlessly with services provided by other sectors.

The introduction of co-commissioning drew much opposition, especially in terms of overcoming real and perceived conflicts of interest associated with GPs commissioning or contracting themselves.[18] The key concerns were around performance management of the core GP contract of CCG members, with powers to issue breach notices and terminate contracts. To mitigate concerns over an increased risk of conflicts of interest, NHSE published a statutory guidance on conflicts of interest.[19–21] Additionally, CCGs are required to establish a Primary Care Commissioning Committee,[14] which is a corporate decision-making committee. This committee is separate from the main CCG Board and has a non-clinician voting majority.

Initially, there was no clear expectation for CCGs to take on co-commissioning. One year following the policy implementation (in October 2015), NHSE 'encouraged' all CCGs to consider applying to take on co-commissioning responsibilities by November 2015.[22] By 2016/2017, 115 CCGs (out of 209) had moved towards delegated arrangements. The government plans to extend the delegation of commissioning responsibility to CCGs to include specialised commissioning and aligning commissioning intentions for NHS, social care and public health services.

## METHOD

We undertook an exploratory approach, combining data from interviews and national telephone surveys, with analysis of policy documents and case studies in four CCGs.[23] The ethnographic data provides rich and real insights into a complex set of issues and the national surveys complemented the case studies, giving an indication of the generalisability of the findings and a sense of change over time. Data were analysed using thematic content analysis.

We started by identifying the official aspirations and 'programme theories'[24] motivating the policy on primary care co-commissioning. This involved interviews with senior policymakers (n=6) who had played a role in the development of co-commissioning policy. We also undertook an in-depth analysis of the main policy documents related to primary care commissioning. These theories were subsequently tested in our case studies. In parallel to the interviews, we created a database of CCGs to explore the uptake of co-commissioning nationally (April to May 2015). This was done by reviewing CCGs' application documents as provided by NHSE with CCGs' agreement (n=147).

From the database created, we selected a sample of CCGs to target for two telephone surveys. Out of 209 CCGs, we selected a sample of 104 CCGs. To achieve a maximum variation sampling, our criteria included; the level of co-commissioning responsibility, the regional

team the CCG belonged to, size of CCG, urban vs rural CCG and those undertaking collaborative commissioning with neighbouring CCG. The first telephone survey (n=49) was conducted at 1 year following the policy announcement (June to August 2015). We repeated the survey (n=21) at 2 years following the policy announcement (August to October 2016). We contacted the same sample of CCGs to ask about the development of co-commissioning locally, to see whether their initial objectives for involvement were the same, and whether the CCG had realised any benefits from the new responsibility. Job title and roles of the participants varied between CCGs but in general, we interviewed the following people: director/associate director/senior manager for primary care commissioning, director for strategic commissioning, chair of Joint Co-Commissioning Committee, head of primary care and CCG chair/chief officer/accountable officer/medical director/managing director.

Lastly, we conducted case studies[25] in four CCGs nationally (January 2016–April 2017). Cases were chosen to cover different regions, population sizes, contractual mechanisms, new models of care and sustainability and transformation plan areas. Our observations focused mainly on meetings associated with co-commissioning such as the Primary Care Commissioning Committee and its subcommittees or subgroups. We attended a total of 74 meetings (approximately 111 hours of observations) and conducted a total of 42 face-to-face interviews with committee members such as the primary care manager, head of estates, local medical council representative and CCGs' governing body members.

### Patient and public involvement
No patients and public were involved in study.

## RESULTS
### Programme theories
We identified two programme theories underpinning the policy[26]:
1. Integration of budgets and commissioning responsibility with a single commissioner for commissioning primary, community and secondary care for a geographical population. This would allow the shifting of resources between sectors, facilitate the development of a more integrated approach to service provision and provide an environment, which would support the development of integrated organisations. This would then deliver more care outside hospitals and care, which from the patient's perspective would be more integrated, efficient, effective and cheaper.
2. CCGs understand primary care and local needs. Allowing CCGs to commission primary care, alongside other services CCGs were already commissioning, would support the development and implementation of local strategies for service improvement, support innovation in primary care and allow investment in primary care (by allowing resource shifting as outlined

above). This would improve the quality of care, make primary care a more attractive place to work and facilitate recruitment and retention.

### Primary care co-commissioning activities
#### Strategic plans
Our case study CCGs had developed strategic plans which outlined how they were planning to support, enable, strengthen, sustain and/or transform general practice to address the challenges or pressures that the local health systems were facing. The plans were also developed to deliver the aspirations in the FYFV[15] and General Practice Forward View (GPFV)[27] and were used as a basis to develop the place-based strategies for sustainability and transformation partnerships (STPs).[28]

All of the case study CCGs faced increasing demand due to an ageing population and patients with multiple and complex needs, which led to an increased workload. These challenges, along with GP shortages and financial pressures put the local health systems under immense pressure meaning patients did not always receive the quality and standard of care they needed. In their plan, CCGs claimed that these challenges could be addressed by developing a more integrated approach to delivering health and social care services for the local community. The vision was to achieve a people-centred, locally driven, integrated primary care service with general practice at its heart.

Investment and opportunities contained in national and local initiatives were seen as major contributors to enabling CCGs achieving their vision. Investment identified by the CCGs included; the CCGs' core budget, the delegated budget for primary care commissioning, the GP Access Fund,[29] Vanguard funding[15] and the GPFV funding stream.[27]

#### 'New' primary care initiatives
Our case study CCGs were involved in supporting the development of new models of care in their local area. The support that the case study CCGs provided to GP practices was through various 'new' forms of local incentive schemes (also called 'contracts' or 'frameworks'). An impetus for these schemes was the need to redistribute money released from a review of GMS and PMS contracts, which sought to equalise payments to practices providing equivalent services. CCGs were driven by a need to improve access, promote a consistent level of service delivery across the CCG and encourage practices to develop new ways of working. In addition to monies from the existing primary care budget, these 'new' initiatives were funded through the consolidation of existing enhanced services and the wider CCG budget. However, there was limited scope for significant shifting of funds between services because of budgetary pressures, with some CCGs forced to use primary care funds to support secondary care budgets.

Most of the 'new' initiatives introduced by the CCGs in our case study sites were designed to support GP practices

working together or working 'at scale' and to streamline various local incentive schemes into a unified contract to reduce the need to monitor the delivery of multiple contracts. For example, one of our case study sites introduced an all-in-one scheme which replaced all local individual incentive schemes to enable practices to work together to deliver a more consistent level of services and facilitate primary care to engage more effectively with secondary care. Practices were free to collaborate with other practices 'at scale' to deliver the standards. These initiatives were described as a means to 'to drive forward collaboration' (GP ID16) and a 'strategic step towards budget delegation and (development of new models of care)' (Primary Care Commissioning Committee meeting site 1). In another case study site, the CCG decided to introduce a variety of schemes to achieve the same objective, which was to streamline existing schemes into a unified contract.

### Quality and outcomes framework

A recent review of the national quality pay for performance scheme the quality and outcomes framework (QoF) found that most of the indicators were unlikely to promote the aims of the FYFV relevant to primary care, including integrated and patient-centred care leading some CCGs with new models of care to discontinue QoF.[30] Nevertheless, all of our case study CCGs continued with QoF, despite the power to negotiate a local alternative. It was argued that developing a local QOF would not be an easy process, a view informed by the experience of developing the local incentive contracts, and CCGs felt their members had little appetite for this type of change.

### Legacy work

In our observations in case study sites, we found that much of the committees' time was taken up in dealing with legacy issues inherited from NHSE. These were mostly in the area of primary care estates and issues to do with APMS contracts.

Estates proved to be a particular issue, largely because of a significant loss of expertise in this area following the creation of CCGs. Because CCGs initially had no role in primary care commissioning, staff with expertise in primary care property and finance left the organisation. However, one of our case study CCGs which had some resource to employ a head of estates argued that this had enabled the CCG to make efficiency gains, for example by exploring the possibilities of moving the CCG headquarters to a cheaper location.

Legislation, particularly around lease holding for buildings built under the private finance initiative contracts proved unclear. Our respondents told us that they felt that NHSE had provided limited guidance in dealing with sometimes fraught issues relating to property management. More widely, the financing and management of primary care in England mean that individual GP practices retained responsibility for decisions about investment in the development of buildings. This means that

CCGs—which had overall responsibility for the strategic direction of primary care services—had limited levers with which to influence the development of buildings and facilities.

CCGs also had to deal with legacy issues associated with APMS contracts. For example, in one of our sites they were unable to obtain a copy of the APMS contract supposedly governing the provision of services by non-traditional GP providers, while another CCG found that a supposedly time-limited APMS contract had been drawn up with no end date.

## Wider national initiatives
### New models of care

We found mixed views among our respondents as to what new collaborative service models would mean for the future of primary care and the impact they could have on CCGs and their members. Our case study CCGs could see the opportunities arising from implementing new models of care with some enthusiastic about the direction of travel. However, others were more pessimistic, largely due to resistance to change or change apathy:

> I think there's some hesitation by our GPs. So, we're a member organisation. I think there's some worry over what it means with our GP colleagues across the patch and …I think there's a cohort of people who see it as an opportunity to shape the future and then there's a cohort of people who think, you know, it's concerning about the future of general practice. [Independent GP ID9]

Some were concerned about the potential difficulties of bringing people together through the new models of care who would not necessarily choose to work together or have no previous history of working together:

> You can't just throw half a dozen professionals in a room and just expect them to work together, because integration is more than co-location in my view. [Independent GP ID9]

> […] you can't force primary care, individual practices to work with each other, if they haven't got a history of a relationship or some trust, so there's lots of work that needs to be done. [Manager ID19]

Hence collaborative working was something that could not be taken for granted but must be worked on and facilitated. Factors with the potential to facilitate collaborative working included having a trusted peer who could convince their GP colleagues of the merits of collaborating and adopting a slow and iterative approach when introducing changes affecting practices. Notably, we found that, while CCGs were supportive of new models of care, the incentive schemes discussed earlier focused on primary care-specific activity using existing repurposed budgets, rather than any shifting of resources between service sectors. In particular, we saw no evidence of shifting resources from secondary or community services

to primary care, and no pooling of budgets between these sectors.

## Sustainability and transformation partnerships

Interviewees described how they grappled with understanding which services could be planned and commissioned locally, on a 'place' basis, and which could be provided on a 'wider footprint' such as the STP. The deliberation seemed to couch around population size, geographical footprint, the relationship with local hospitals and LAs and the current system in place. Interviewees felt that the local level was where primary care should be commissioned and delivered:

so because primary care is commissioned on a very local footprint, when you talk about [STP area], I don't think you talk about primary care that much, because you don't need to do that on a [STP area] basis, so why do we need, you know, because inevitably, I think people will talk about it on the basis of, I'm interested in it, I'm passionate about it, therefore, I like to talk about it, but if talking about it on a [STP area] basis means we're all going to do it the same across [STP area] while there's somebody else called head of [STP area] going to make the decision on my behalf, then I don't' think I want to talk about primary care anymore, because I'm deciding how we're going to do it in [CCG area]. […] So it inevitably gets mentioned in STPs and stuff, but I don't think the STP is a construct that really lends itself to commission primary care. [Manager ID42]

One of the rationales behind the transfer of primary care commissioning responsibilities from NHSE to CCGs was to support the development and implementation of 'local' strategies supported by 'local' knowledge.[26] Interviewees expressed a concern that STP footprints were too large to truly take account of the local needs of primary care services. They described the process of linking their own locally based primary care plan with STPs as akin to 'knitting' or a 'jigsaw'. A key challenge for CCGs was to connect the two:

So, my view, I suppose our view on that [primary care] is actually that we needed a local plan about what all of that looks like for us and we'll feed that to the STP, so it's a two-way process, the STPs have come up with their high level work plans and they will have leads working on those. […] The challenge for us will be in keeping connectivity between the two so that, you know, they are aligned, otherwise we could end up with an STP plan that says we are going to do X,Y, Z and local plan that says we are going to do A,B,C. Neither the two will meet and nothing will get delivered and the challenge that we have got to do as a system, it means we can't afford to do that. [Manager ID19]

One of the main challenges identified by our respondents was the fact that STPs have no legal basis,

and hence there were concerns expressed about their accountability and governance. However, there was also a recognition that some sort of regional coordination function was required if integrated services were to be delivered:

But PCT clusters were probably in the right scale, and it's no great surprise to me that the three STPs that we've got are the same as the three PCT clusters that we broke up to create CCGs. […] I think it's because there is just a sensible way to organise the commissioning of health and social care really. You have to be able to work at a certain scale, you have to be able to commission a range of services across the spectrum of the care pathway, you have to be able to work in partnership with local government to make it a success. So, you know, however you chop it up, basically eventually, you end up coming back to those design principles; and when you apply those you end up with PCT clusters, or STP or, you know. So it's a kind of unassailable logic really. [CCG Accountable Officer ID18]

## Work on integration with the LAs

CCGs were working with local authority (LAs) on broader portfolios of work, which although not directly part of primary care commissioning, had an impact on general practice. For example, one case study CCG was implementing a new model of care, with an emphasis on integration between health and social care. A priority for the CCG was that primary care was included in the integrated model of working by ensuring that primary care was discussed and understood by the different stakeholders.

The relationship between CCGs and LAs was heavily influenced by austerity and the continual budget reductions that faced LAs. In another case study CCG, there were numerous previous attempts to integrate health and social care. However, at a certain point organisations became protective of their own budgets, which inhibited more integrated ways of working:

A very bitter example of course is health and social care and how for so many years we've not been able to match the two things together, because one is in the hands of the local authority and the other is in the hands of the NHS, the health service. Why have we taken so long to even start talking about bringing these…? And we have tried in the past. Section 75 payments, whatever, groups of people like people with learning difficulties, we've worked with those, because they have a foot in both camps, local authority and health, and we've tried very hard to pool budgets together but then the local politics kicks in. There's a gap in the local authority finances and the NHS is worried that if you give them any money it'll disappear forever and it will never be used for those…[Lay member ID14].

# DISCUSSION

The priority to coordinate and integrate health and social care services has been recognised internationally. One of the ways to do this is by strengthening primary care. In the English NHS, this has been done by the active commissioning of primary care services through the introduction of APMS contract in 2004. The policy has for a number of years emphasised the need for commissioners to move beyond passive allocation of resources to actively considering mechanisms and incentives to ensure that primary care delivery supports policy objectives. The recent delegation of primary care commissioning responsibilities from NHSE to CCGs was intended to give CCGs the ability to shift resources between different sectors in order to support integrated care.[14] However, there is little published evidence about how CCGs are responding to their new primary care commissioning responsibilities. This is the first study to provide such evidence. However, as the study was undertaken in a rapidly changing policy environment, the results represent a snap shot in time of a changing landscape.

In practice we found that, while CCGs and their constituent practices are supportive of the idea of integrated care as evidenced in their strategic plans and were actively using their primary care commissioning power to incentivise and support collaborative working among practices, there was limited evidence of wider moves towards integrated care in the community such as integrated budgets between primary and community services or between health and social care, or shifting of resources away from secondary care to support services outside hospitals. The focus of co-commissioning activities was rather on incentivising local practices to proactively manage patients to reduce the use of secondary care services and to work together in local groups to achieve this.

While there are many factors at work which may explain this, including a nervousness among GPs about anything which would potentially risk funding being shifted away from primary care to support other community-based services, part of the explanation may also lie in the fact that the commissioning of primary care services by CCGs is a policy 'workaround'. The legislation establishes NHSE as the statutory authority commissioning primary care services. The delegation of this function to CCGs does not remove this statutory responsibility, and introduces significant issues in relation to potential conflicts of interest.[18] As a result, structures have been put in place within CCGs to separate the work of commissioning primary care services from the wider work of the CCG in commissioning secondary and community services, with Primary Care Co-Commissioning Committees constituted as separate decision-making bodies which are required to have a non-GP majority. This separation means that there is limited opportunity for primary care services to be considered in the wider context of the CCG's strategy, and thus limited scope for a truly population-focused approach to integrated care.

To alleviate these problems and to support a broader integration agenda, it has been suggested that integration has to be pursued at different levels within a system,[3] with local collaboration among GP practices providing one element in wider new models of care across a broader footprint. Thus, for example, it has been suggested that GP 'networks' could act together as a provider entity within a broader community-based integrated organisation, and that a number of these new provider groupings or alliances could work together across a large footprint to form an 'integrated system'. However, in our study, we found a disconnect between locally based primary care and the wider system. Our study identifies that one of the major challenges to integrate care vertically or 'knitting' the locally-based primary care plan with regional plans (as embodied in STPs) is the lack of knowledge and expertise in the field of primary care at the STP level. Integrated care requires detailed local work to build trust and develop context-specific mechanisms to work across boundaries. While primary care commissioning by CCGs seems to be supporting local collaborations between practices, there is some way to go before this is translated into broader integration initiatives across wider footprints.

**Acknowledgements** We are grateful to our participants who were very generous in allowing us access to their organisations. We would also like to thank Dr Donna Bramwell and Dr Oz Gore who undertook the fieldwork and contributed to data collection, data analysis and final report.

**Contributors** IM, KC, VM, LWG: met the criteria for authorship and contributed to the drafting, revision and finalisation of this paper. IM, KC: drafted the initial version of the manuscript.

**Funding** This work was supported via the Department of Health funded Policy Research Unit in Commissioning and the Healthcare System (grant no 101/0001).

**Disclaimer** The views expressed here are those of the researchers and do not reflect the position of the Department of Health.

**Competing interests** None declared.

**Patient consent for publication** Not required.

**Ethics approval** The study received ethical approval from the University of Manchester's Research Ethics Committee (ref 11104). Verbal agreement and in some cases, signed confidentiality agreement were obtained from research sites. Following agreement from the sites, agreement was also sought from the chair of meetings observed. Written consents were obtained for interviews, and oral consents were obtained for surveys.

**Provenance and peer review** Not commissioned; externally peer reviewed.

**Data sharing statement** There are no additional data available.

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
