## [Reviewer comments · BMJ Open]

ARTICLE DETAILS

TITLE (PROVISIONAL)	Achieving integrated care through commissioning of primary care services in the English NHS: a qualitative analysis.
AUTHORS	McDermott, Imelda; Checkland, Kath; Moran, Valerie; Warwick-Giles, Lynsey

VERSION 1 - REVIEW

REVIEWER	David Wyatt King's College London
REVIEW RETURNED	08-Dec-2018

GENERAL COMMENTS	Thank you for the opportunity to review this interesting paper about the changing primary care commissioning processes as NHS England attempts to move towards an integrated care system. Providing a detailed policy background and draw on a substantial amount of data, the authors present a comprehensive paper on a shifting NHS. It is wholly relevant to the BMJ Open's readership and adds to our knowledge about health policy, shifting health service practices as well as contributing to wider discussions on the management of change in complex systems. With a few minor amendments, I believe this paper would be ready for publication. These are listed by section below and I hope they are useful to the authors. Methods: the authors have used a lot of data in their study but I was unable to grasp the details of the methods they used. For example, they mention their sampling criteria for their telephone survey (pg 7, line 33), but more is needed to make sense of it. For example, how did they use these criteria? Were they aiming for a maximum variant sample or something else? Similarly, how were the four CCGs selected? The authors mention that they "drew out programme theories" and reference Weiss (1997). A sentence or two explaining what this involves would make this more accessible and allow the reader to understand their process better. BMJ Open ask me to specifically comment on research ethics and consent. This is not mentioned in the paper at present (Although I acknowledge the authors received ethical approval from Manchester). Please could the authors add a sentence
--

	documenting how consent was obtained for the research sites, interviews and surveys. Results: In the section, “Primary Care Co-Commissioning Activities”, the authors draw on their case study data. Although making interesting points, the section doesn’t really provide the detail I’d expect from their qualitative data and analysis. At present, case study data (except for one point on Pg 9 line 46), appear homogenous. Were there any differences that they could draw out and add to any of the subsection here to acknowledge some of the variation in practices (as well as the areas of overlap)? How do the different CCGs deal with the same issues but in slightly different ways? Discussion: On Pg 12 the authors explain the way that co-commissioning takes place within the CCG. From observing the co-commissioning meetings, are the authors able to comment on this from their data itself? This seems a really important point about the way the structure of co-commissioning practices inhibits the aim of integrated care. If the authors can evidence this from their data and add to the Results section, this would further strengthen this paper. General point: This is extremely minor. There are a lot of acronyms used in this paper. While I understand that some are necessary, I wondered whether it was possible to reduce these down – For example, GMS and PMS occur twice and it might be better to just use them in full. This would make this paper more accessible to an international audience.
--	--

REVIEWER	Dr Raheelah Ahmad Imperial College London
REVIEW RETURNED	09-Dec-2018

GENERAL COMMENTS	The comments are provided as they arose as I read the paper. So, whilst some information appears later – the reader is unclear until much later. The authors may therefore consider revisions with this in mind. The authors investigate a really important issue and the paper has potential to make a much needed contribution. Although the following point would normally be highlighted under minor revisions, the extraordinary use of NHS centric terminology and acronyms need attention. Sorting this out will improve accessibility to the key learning (which itself is not clear). Title: Achieving integrated care through primary care co-commissioning in the English NHS: a qualitative analysis. Not sure that co-commissioning needs to be in the title – it’s a very particular structure and process which is described in the text but
--

don't think that it is helpful for the international readership - wonder if something more widely understood may be appropriate?

Abstract

Objectives

Not clear what is going to be analysed – 'this recent development' or the change in structures and processes for commissioning as a result of the recent developments? Or some measurable impact in terms of the nature of the commissioned services?

Design

Being more specific about the design than 'exploratory approach' would be helpful if others want to replicate this study – is it an exploratory case study? Or qualitative observational study? Prospective? Retrospective?

The methods for data collection are clearer, but if word count can accommodate then please say what the 'case studies' were of.
(line 17)

I cannot tell from your abstract how you analysed the data. Needs a line.

Setting /participants

Multi-method approach to be commended.

Sample of CCG staff is strong.

Who were the national survey participants?

Same comment about 'co-commissioning meeting' as for title.

Suggested change: "We reviewed 147 CCG application documents" to "We reviewed all 147 CCG applications in (add years/period)" – it is strong that you reviewed all of the applications (not mentioned here) but is in the highlights.

Results and conclusion – because of the use of the very English/NHS centric terminology I don't think the reader would learn much from these two sections.

Also giving an example or two of the 'mixed' findings is needed – as this sounds far too vague and non-committal.

With the suggested revisions the aim and methods will be strong. I am intrigued to read the rest of the manuscript to see what you actually found. But I should know this by now of course having read the abstract – but I don't really.

In the study summary all your highlights are about methodology. Worth thinking about the findings/learning too.

Introduction

The first part of the introduction is a bit confusing with lots of terminology and probably more acronyms than you need to use.

The section then moves on to the history and current context. I think these sections could be better streamlined. For example, the section HISTORY OF COMMISSIONING OF PRIMARY CARE SERVICES IN ENGLAND could be better presented as a figure/timeline with text.

Methods

The description of the case studies could be strengthened with references. It would also be helpful to know in which order the different data sources were collected and analysed. Appears like a collection of activities currently which I am sure is not the case.

You presumably used these sources for triangulation, depth analysis, and assume that collection and analysis was iterative.

Did you make any cross-case comparisons?

	I have found references in the methods section to help me understand the approach used. There is one reference (Weiss) about programme theories but nothing to explain the relevance of programme theories in the text. Earlier reference is made to Valentijn et al. but I see no use of the framework for analysis – which would have been one possible and useful way of analysis. Results This is the weakest part of the paper and it is not clear why, given the richness of the data that must have collected. There are broad headings but no conceptual themes (as one would expect in a qualitative analysis) It is then difficult to know why the conclusions made are the ones picked out. Minor comments: Not clear why single inverted commas have been used, for example: page 4: ‘integrated care’; ‘organisational mechanism’ Page 6: ‘new powers’ (line 45) Please review throughout the manuscript Page 4: Why is commissioning in italics (line 18)?
--	---

VERSION 1 – AUTHOR RESPONSE

5. Reviewer 1 asked us to add a sentence documenting how consent was obtained. This has been added in the revised text on p.14 (under the heading ‘Ethical Approval’).

6. Both reviewers suggested making the richness of the data collected clearer in the Results section. We have added additional texts throughout this section, where appropriate, to clearly indicate variations between case study sites. Findings are also referred to in terms of whether what we observed occurred in ‘one of our case study sites’ or ‘all of the case study CCGs’, and whether there are ‘mixed views’ amongst respondents in our interviews.

7. Reviewer 1 made an interesting point about how the structure of co-commissioning inhibits the aim of integrated care. We have added a sentence describing the structure of primary care co-commissioning committees in the ‘current policy context’ section on p.7.

8. Reviewer 2 argued that it is not clear how the Results link to the Conclusions made. We argue that the Results section is clearly divided into two main conceptual themes (with sub-themes under each). The conclusion drawn, based on these two themes, is that the activities undertaken under primary care co-commissioning (the first main theme) are mainly focused on incentivising and supporting collaborative working amongst practices as evidenced in their strategic plans, ‘new’ primary care initiatives etc (the sub-themes under the heading ‘primary care co-commissioning activities’). We found a disconnect between these locally-based activities and plans with the wider national initiatives to achieving integrated care (the second main theme).

9. Reviewers 1 and 2 suggested reducing the use of very English/NHS centric terminology to make the paper more accessible to an international audience. We have addressed this by using fewer acronyms throughout the paper.

10. Reviewer 2 asked us to consider revising the title to something more widely understood for international readership. We have revised the title as suggested by replacing the term 'co-commissioning' with 'commissioning of primary care services'.

11. Reviewer 2 noted that the relevance of programme theories was not explained in the text. We have added text to the Results section on page 8 to address this point.

12. Reviewer 2 asked us to be more specific in using the term 'exploratory approach'. We argue that this term is appropriate and specific about the design of the study. The specific methods of data collection (from various sources) and analysis are specified in the Abstract and further elaborated in the Method section of the paper.

13. Reviewer 2 asked us to give an example or two of the mixed findings in the Abstract. We argue that the Abstract is not the appropriate section to provide examples and has a strict word limit. Any examples are provided in the Results section of the paper.

14. Reviewer 2 suggested that the History section could be better presented as a figure/timeline with text. We argue that the use of figure/timeline would reduce the nuance needed to provide the necessary context for the paper, which is needed for international audience.

15. Reviewer 2 asked about the use of single inverted commas. For the term 'integrated care', we use this in the Introduction to indicate our awareness of the different meanings that may be associated with this term. For the term 'organisational mechanism', this is used to indicate that we are using this term as defined by Calnan et al. For the term 'new powers', this is used to show that this is the term used by Simon Stevens in his speech.

16. The Editor suggested formatting amendments to include Data Sharing Statement and Patient and Public Involvement. These have been added in the revised text on p. 14 and p.8.

VERSION 2 – REVIEW

REVIEWER	David Wyatt King's College London United Kingdom
REVIEW RETURNED	25-Jan-2019

GENERAL COMMENTS	Thank you for the opportunity to review this paper again. The authors have addressed the points from my previous review.
---

REVIEWER	Dr Raheelah Ahmad Imperial College London
REVIEW RETURNED	25-Jan-2019

GENERAL COMMENTS	Thank you for the clear revisions made to the manuscript.
---